

# Re-positive of SARS-CoV-2 test is common in COVID-19 patients after hospital discharge. Data from high standard post-discharge quarantined patients in Macao SAR, China

Chi Leong Wong[1,*], Sao Kuan Lei[1,*], Chin Ion Lei[2], Iek Long Lo[3], Chong Lam[4] and Iek Hou Leong[4]

[1] Macao Academy of Medicine, Centro Hospitalar Conde de São Januário, Health Bureau, Macao SAR, China
[2] Department of Medicine, Centro Hospitalar Conde de São Januário, Health Bureau, Macao SAR, China
[3] Department of Respiratory Medicine, Centro Hospitalar Conde de São Januário, Health Bureau, Macao SAR, China
[4] Center for Disease Control and Prevention, Health Bureau, Macao SAR, China
[*] These authors contributed equally to this work.

Corresponding author
Chin Ion Lei, cilei@ssm.gov.mo

## ABSTRACT

During the pandemic of COVID-19, Macau faces tremendous pressure because it is a famous gambling and tourism city with the world's highest population density. The Macau government implemented decisive public health intervention to control the transmission of COVID-19, and there were only two independent outbreaks in Macau. In the second outbreak, all 35 cases were infected in foreign countries. They were quarantined in airborne infection isolation rooms for at least 14 days with reverse transcription-polymerase chain reaction (RT-PCR) tests after hospital discharge. Twelve (34.3%) of them had re-positive SARS-CoV-2 test results, and none of them presented any COVID-19 signs or symptoms during the entire quarantine period. In this study, the re-positive patients were more likely to be diagnosed in the early stage of the disease with a longer hospital stay. Virus re-infection is impossible in this high standard isolation setting, and reactivation is also unlikely, so that residual virus nucleic acid should be the possible reason for this phenomenon. Due to limited data about the risk of re-positive patients, it is better to quarantine patients after discharge for a prolonged period with repeat RT-PCR tests to minimize the community's potential risk, particularly in the regions with relative plenty of resources and low community infection rate such as Macau.

## INTRODUCTION

In December 2019, a series of pneumonia cases of novel coronavirus infection emerged in Wuhan city of Hubei province in China. Then, the disease spread quickly and widely worldwide and was named COVID-19 (*Li et al., 2020*). As a rising number of the issue, the WHO declared the outbreak as a Public Health Emergency of International Concern on

30 Jan 2020 (*World Health Organization, 2020a*). Up to early November, the accumulative number of confirmed cases has reached over 46 million worldwide, and it continues to rise (*World Health Organization, 2020b*). Some studies even found the Spike D614G mutation of the virus with increased infectivity in other countries while comparing the virus samples in China (*Korber et al., 2020*). As a famous gambling and tourism city with the highest population density globally, Macau faces massive stress from overseas and local populations during the coronavirus pandemic. From our point of view, Macau fights against COVID-19 quite successfully. There were only 45 confirmed cases in Macau's two irrelevant outbreaks before late June, without local transmission. All ten patients in the first outbreak from 22 Jan to 4 Feb were from Mainland China, while the 35 cases in the second episode from 15 Mar to 8 Apr came from foreign countries. After that, there were no more new cases for over two months (*Macao SAR Government Portal, 2020*). The Macau Government's tremendous effort contributed to the great success in this fighting.

We made early upgrades to our port policy to a 14-d quarantine with at least twice RT-PCR tests of novel coronavirus for all people who came from other countries since March (*Lio et al., 2020*). On the other hand, we had a loose community testing policy, making RT-PCR tests accessible to everyone whenever they requested. Under these aggressive public health strategies, we could safely confirm all infected cases in the second outbreak.

Meanwhile, studies reported SARS-CoV-2 re-positive cases after the patients fulfilled discharge criteria (*Lan et al., 2020*; *Yuan et al., 2020*; *Zou et al., 2020*; *Wang et al., 2020*). It raised concerns about the virus's re-infection or reactivation, regardless of whether there was no substantial evidence about their infectivity. Viewing this, we quarantined all patients who fulfilled discharge criteria in airborne infection isolation rooms, for at least 14 days, with RT-PCR test on day 10 and day 13, before they were back to the Macau community. The re-positive rate in academic reports was 3.3% to 21.4% (*Yuan et al., 2020*; *Zou et al., 2020*; *Wang et al., 2020*; *Kang, 2020*; *Xiao, Tong & Zhang, 2020*; *Justin et al., 2020*). Most of their patients became infected in China. Some were home isolated while some used inconsistent novel coronavirus test methods and other surveys represented only partial cases in a city. Here, we reported the re-positive rate of all COVID-19 patients in Macau during the second outbreak. All of them were infected outside of China and observed under a high standard quarantine location with repeated nasopharyngeal swab RT-PCR tests after hospital discharge.

## MATERIALS & METHODS

### Study design

This retrospective study included all COVID-19 confirmed cases ($n = 35$) in Macau during the second independent outbreak which was from 15 Mar, 2020 to 8 Apr, 2020. They were from foreign countries and were admitted to Centro Hospitalar Conde de São Januário (CHCSJ), Macau SAR, China. We discharged the patients when they fulfilled the criteria of the 7th edition from the National Health Commission of China, which included obvious improved symptoms and pulmonary imaging and at least twice consecutively negative RT-PCR tests on nasopharyngeal samples (*National Health Commission &*

*National Administration of Traditional Chinese Medicine, 2020*). Then, the patients were transferred to a standard isolation ward for at least 14 days for post-discharge quarantine. Experienced medical staff closely observed them and performed two times RT-PCR assay on their nasopharyngeal specimen on day 10 and day 13 of quarantine. The negative group was the cases with negative results in all RT-PCR tests, while re-positive patients were the ones who had at least one positive RT-PCR test during the quarantine. Before returning to the community, the re-positive subjects had to repeat the test every other day until three consecutive negative results. The Hospital Medical Ethical Committee of Centro Hospitalar Conde de São Januário, Macao SAR, China approved this study (Ethical Application Ref: 0044/MEC/N/2020). The committee waived the requirement for patient consent.

## Data collection

We collected the data from both electric and written medical records. It consisted of demographic data, epidemiological data, clinical data, including severity classifications, length of stay, symptoms, signs, laboratory results, radiological results, main treatment strategies, and post-discharge quarantine data.

## Standard isolation ward

There were 60 airborne infection isolation rooms (AIIR) in our quarantine center with up to two beds per room, but only minors could share space with the family members. Each room had an anteroom, electric self-closing doors with an interlocking system, clean to dirty airflow, negative pressure monitor, with at least 12 air exchanges per hour, and high-efficiency particulate air filtration, which fulfilled American Centers for Disease Control and Prevention suggestion (*Centers for Disease Control, 2003*). Furthermore, qualified doctors and nurses were working in the ward 24 hours a day. All staff used personal protective equipment (PPE), including gloves, isolation gown, N95 mask, face shield, headgear, and foot cover when they entered AIIR, and discarded them properly at once when they left the room to avoid cross-infection.

## Reverse Transcriptase-Polymerase Chain Reaction (RT-PCR) test for SARS-CoV-2

Nasopharyngeal swab (NPS) samples were tested using a commercial SARS-CoV-2 ORF1ab/N Gene Nucleic acid detection kit (BioGerm, China) following the manufacturer's instructions. A cycle threshold value (Ct-value) less than or equal to 35 was defined as a positive test result.

## Statistical analysis

SPSS ver. 25.0 (SPSS Inc., Chicago, IL, USA) conducted the statistical analyses. Continuous data were presented as median $\pm$ interquartile range (IQR); categorical variables were presented as frequency/percentage. For continuous variables, the Mann–Whitney U test was used for intergroup comparisons among groups with skewed distributions. For categorical variables, the chi-square test or Fisher's exact test was used. A two-sided significance level of 0.05 was used to evaluate statistical significance.

## RESULTS

### Re-positive rate and clinical characteristics of the re-positive group

There were 35 confirmed cases in this study, and 12 (34.3%, 95%CI [21.5%–49.3%]) were found to be re-positive during the post-discharge quarantine in AIIR. Both males and females had six people, and the median age was 28.5 years old (IQR: 18.3–42.3). Serum IgG antiviral antibody was positive in all patients before the quarantine. The median days between the re-positive NPS sample and IgG antibody detection was 30 days (IQR: 19.3–44). Furthermore, none of the cases had any signs or symptoms of COVID-19 during the entire quarantine period.

### Risk factor analysis

Age and gender: The median age of the negative group was 28 years old (IQR: 20–44), like the re-positive group. There were seventeen males and six females in the negative group, with a 26.1% re-positive rate in male patients and 50% in females. There were no statistically significant differences in age or gender between the re-positive and negative groups (Table 1, $p > 0.05$).

Nationality and travel history: Most of the patients were Chinese ($n = 23$, 65.7%) and back from England ($n = 16$, 45.7%). The re-positive rate in Filipino ($n = 3$) and Korean ($n = 1$) were 100% while Chinese was 34.8%, Indonesian ($n = 3$) and Portuguese ($n = 3$) was 0%, closed to being statistically significant (Table 1, $p = 0.066$). For travel history, re-positive rate had no statistically significant differences between different groups (Table 1, $p > 0.05$).

Clinical condition: Signs and symptoms, clinical classification, past medical history, laboratory test, computed tomography, and therapeutic schedules were compared between the two groups. None of them had significant differences (Tables 1 and 2, $p > 0.05$). Nevertheless, we noticed that those clinically classified as asymptomatic types had no re-positive results. Simultaneously, those with some signs or symptoms before hospitalization were more likely to be re-positive (Table 1, 39.3% vs. 14.3%).

Timeline of disease progression: We analyzed the duration between the onset of symptoms or signs (S/S), diagnosis, and the date of hospital discharge criteria fulfilled. There were seven patients without any S/S before admission. In the re-positive group, the median days from S/S onset to diagnosis was shorter (2 days vs. 4 days) while the median days from S/S onset to hospital discharge and from diagnosis to hospital discharge were longer (43 days vs. 35 days & 38 days vs. 27 days). However, no significant difference was found in these parameters between the two groups (Table 3, $p > 0.05$).

## DISCUSSION

One-third of patients had re-positive SARS-CoV-2 test results in our study, which is higher than other studies (*Yuan et al., 2020*; *Zou et al., 2020*; *Wang et al., 2020*; *Kang, 2020*; *Xiao, Tong & Zhang, 2020*; *Justin et al., 2020*). Under the outstanding public health strategies and an ideal environment in Macau, we could evaluate the re-positive rate of all the patients quarantined in a high standard location with RT-PCR tests after hospital discharge within

**Table 1  The clinical characteristics of re-positive group and negative group.**

| Clinical characteristics | Groups | | χ²/Z value | P value |
| --- | --- | --- | --- | --- |
| | Re-positive group | Negative group | | |
| **Age**, median (IQR) | 28.5 (18.3–42.3) | 28 (20.0–44.0) | −0.626 | 0.542 |
| **Gender**, No. (%) | | | | |
| Male (n = 23) | 6 (26.1) | 17 (73.9) | – | 0.149 |
| Female (n = 12) | 6 (50.0) | 6 (50.0) | | |
| **Nationality**, No. (%) | | | | |
| Chinese (n = 23) | 8 (34.8) | 15 (65.2) | 11.843 | 0.066 |
| Filipino (n = 3) | 3 (100) | 0 (0) | | |
| Indonesian (n = 3) | 0 (0) | 3 (100) | | |
| Portuguese (n = 3) | 0 (0) | 3 (100) | | |
| Australian (n = 1) | 0 (0) | 1 (100) | | |
| Korean (n = 1) | 1 (100) | 0 (0) | | |
| Spanish (n = 1) | 0 (0) | 1 (100) | | |
| **Travel history**, No. (%) | | | | |
| England (n = 16) | 5 (31.3) | 11 (68.8) | 8.129 | 0.421 |
| Philippines (n = 5) | 3 (60.0) | 2 (40.0) | | |
| Portugal (n = 4) | 1 (25.0) | 3 (75.0) | | |
| America (n = 3) | 2 (66.7) | 1 (33.3) | | |
| Indonesia (n = 3) | 0 (0) | 3 (100) | | |
| Cambodia (n = 1) | 0 (0) | 1 (100) | | |
| Ireland (n = 1) | 1 (100) | 0 (0) | | |
| Spain (n = 1) | 0 (0) | 1 (100) | | |
| Thailand (n = 1) | 0 (0) | 1 (100) | | |
| **S/S before hospitalization**, No. (%) | | | | |
| Yes (n = 28) | 11 (39.3) | 17 (60.7) | – | 0.217 |
| No (n = 7) | 1 (14.3) | 6 (85.7) | | |
| **Classification when admission**, No. (%) | | | | |
| Asymptomatic type (n = 3) | 0 (0) | 3 (100) | 1.712 | 0.425 |
| Mild type (n = 16) | 6 (37.5) | 10 (62.5) | | |
| Moderate type (n = 16) | 6 (37.5) | 10 (62.5) | | |
| **Classification during hospitalization**, No. (%) | | | | |
| Asymptomatic type (n = 3) | 0 (0) | 3 (100) | 3.061 | 0.382 |
| Mild type (n = 12) | 6 (50.0) | 6 (50.0) | | |
| Moderate type (n = 17) | 5 (29.4) | 12 (70.6) | | |
| Severe to critical (n = 3) | 1 (33.3) | 2 (66.7) | | |
| **Past medical history**, No. (%) | | | | |
| Yes (n = 6) | 1 (16.7) | 5 (83.3) | – | 0.311 |
| No (n = 29) | 11 (37.9) | 18 (62.1) | | |
| **Laboratory test**, median (IQR) | | | | |
| Highest CRP | 0.31 (0.11–1.13) | 0.32 (0.05–1.88) | −0.243 | 0.817 |
| Highest PCT | 0.05 (0.03–0.06) | 0.03 (0.03–0.06) | −1.078 | 0.290 |
| **CT when admission**, No. (%) | | | | |
| Normal (n = 19) | 6 (31.6) | 13 (68.4) | – | 0.495 |
| Pneumonia (n = 16) | 6 (37.5) | 10 (62.5) | | |

**Table 2  The main treatment used in re-positive group and negative group.**

| Main treatment | Groups | | P value |
|---|---|---|---|
| | Re-positive group | Negative group | |
| **Lopinavir/ritonavir**, No. (%) | | | |
| 2 weeks ($n = 13$) | 3 (23.1) | 10 (76.9) | 0.243 |
| 3 weeks ($n = 22$) | 9 (40.9) | 13 (59.1) | |
| **Interferon**, No. (%) | | | |
| Used ($n = 12$) | 4 (33.3) | 8 (66.7) | 0.618 |
| Not used ($n = 23$) | 8 (34.8) | 15 (65.2) | |
| **Azithromycin**, No. (%) | | | |
| Used ($n = 32$) | 12 (37.5) | 20 (62.5) | 0.271 |
| Not used ($n = 3$) | 0 (0) | 3 (100) | |
| **Levofloxacin**, No. (%) | | | |
| Used ($n = 11$) | 4 (36.4) | 7 (63.6) | 0.576 |
| Not used ($n = 24$) | 8 (33.3) | 16 (66.7) | |
| **Methylprednisolone**, No. (%) | | | |
| Used ($n = 2$) | 1 (50.0) | 1 (50.0) | 0.545 |
| Not used ($n = 33$) | 11 (33.3) | 22 (66.7) | |
| **Hydroxychloroquine**, No. (%) | | | |
| Used ($n = 7$) | 2 (28.6) | 5 (71.4) | 0.547 |
| Not used ($n = 28$) | 10 (35.7) | 18 (64.3) | |

**Table 3  Timeline of disease progression in re-positive group and negative group.**

| Timeline of disease progression, days | Groups | | Z value | P value |
|---|---|---|---|---|
| | Re-positive group | Negative group | | |
| **S/S onset to diagnosis**, median (IQR) | 2.0 (1.0–5.0) | 4.0 (1.5–11.5) | −1.445 | 0.154 |
| **S/S onset to hospital discharge**, median (IQR) | 43.0 (26.0–55.0) | 35.0 (27.5–41.5) | −1.177 | 0.249 |
| **Diagnosis to hospital discharge**, median (IQR) | 38.0 (25.3–50.5) | 27.0 (20.0–38.0) | −1.827 | 0.069 |

one independent outbreak. After all the serious intervention and observation, we could conclude that a significant proportion of discharged patients in Macau were carriers of the virus nucleic acid. Because they were all imported cases in our study, we suspected this phenomenon might be shared globally.

The re-positive rate in our study was as high as 34.3%. A more significant number of tests may be one possible reason. We performed twice tests while the other reported countries only had once (*Zou et al., 2020*; *Wang et al., 2020*; *Kang, 2020*; *Xiao, Tong & Zhang, 2020*; *Justin et al., 2020*). The probability of obtaining a false-negative result increased with time from symptom onset and could be drastically reduced by repeat testing (*Wikramaratna et al., 2020*). In previous studies, the heterogenic design, case selection, and various sampling sites and frequency led to a relatively doubtful re-positive rate (*Yuan et al., 2020*; *Zou et al., 2020*; *Wang et al., 2020*; *Kang, 2020*; *Xiao, Tong & Zhang, 2020*; *Justin et al., 2020*). In contrast, our result should provide more accurate data since we assessed all cases in an outbreak with a unified approach.

Apart from false-negative results, false-positive tests should be considered another likely reason for the inaccurate re-positive rate. The primary route of false-positive PCR test is contamination, including cross-contamination between specimens or synthetically derived target nucleic acids (*Huggett et al., 2020*). All our specimens were performed in an ISO 15189 accredited medical laboratory. Therefore, the likelihood of contamination should be low, and subsequently the false-positive cases.

Along with the global dispersal of COVID-19, mutant viruses have emerged and lead to alteration of the virus behavior. The Spike protein amino acid change D614G is one of the most concerned SARS-CoV-2 variants, which has become apparent in Europe and rapidly replacing other versions of the virus globally since late February, except in China (*Korber et al., 2020*).Different from most existing research, all cases in this study were from foreign countries and might be infected by the D614G variant. One recent research found that the D614G replacement was associated with higher viral loads (*Volz et al., 2021*). It can be an explanation of the unexpected high re-positive rate.

Our study showed a trend that symptomatic patients were more likely to be re-positive. It also found that the median days from S/S onset to diagnosis was shorter in the re-positive group than the negative group, but the duration from diagnosis to hospital discharge was the opposite. However, their statistical significance could not be evaluated due to the small sample size. This result reflected that re-positive patients were more likely to have an earlier diagnosis and more extended hospitalization than the negative group. In other words, the symptomatic and early diagnosed patients tended to be re-positive during the quarantine period. Can there be a higher virus load in vivo cells in this group of patients, with more pronounced symptoms and signs that make a more straightforward diagnosis? Further study is worth to clear this issue.

Although several studies analyzed the risk factors of recurrent positive SARS-CoV-2 test among discharged patients (*Yuan et al., 2020*; *Zou et al., 2020*; *Wang et al., 2020*; *Xiao, Tong & Zhang, 2020*; *Justin et al., 2020*), there was no unanimous conclusion till the present moment. Some research found the re-positive result related to different risk factors such as age, symptoms, illness severity, laboratory tests, etc. Still, they had the opposite contribution in other reports. Indeed, none of the present studies could discover reliable indicators to forecast the patient's risk of being re-positive SARS-CoV-2. Our analysis did not establish any risk factors in terms of demographic or clinical characteristics, laboratory examination, or treatment strategies. As a high re-positive rate was noted and failure to discover risk factors, should re-positive be a natural pathophysiological process of COVID-19?

Except for laboratory error, the causes of a re-positive SARS-CoV-2 test result after hospital discharge have several assumptions. They are re-infection, reactivation, and residual virus nucleic acid.

Hong Kong experts confirmed the possibility of re-infection in COVID-19 patients in a recent study (*To et al., 2020*). It could be one reason for the re-positive test in the previous academic reports. Unclarified or home isolation strategy could imply a little or some opportunity for the people to be exposed to the infection source again. Our research quarantined all patients in airborne infection isolation rooms and under management by professional medical staff with suitable personal protective equipment. Meanwhile, there

was no infection report in medical staff and no local infection case in Macau. As a result, we can exclude the chance of re-infection in this high standard environment.

Reactivation is unlikely since all our re-positive cases were asymptomatic during post-discharge quarantine, and they already had IgG antiviral antibodies in serum samples before discharge. After deducting the above hypotheses, residual virus nucleic acid in vivo cell is a possible cause of the re-positive phenomenon.

Liu WD and colleagues (*2020*) reported a COVID-19 case with prolonged virus shedding even after seroconversion. It implied that the SARS-CoV-2 could exist in human cells in vivo for an extended period. On the other hand, Justin W and colleagues observed the oscillation of positive and negative SARS-CoV-2 NPS test results in the patients, which related to a fluctuating cycle threshold (Ct) value. Therefore, we presume virus load would descend along with the disease's recovery, with a relative up and down pattern rather than in a straight line. When infected cells were intermittently shedding at the level above and below the detection limit, conversion between positive and negative results would appear. Once we conducted the test at the point of fewer shedding cells, the result would be negative, and the patient would fulfill the discharge criteria. Actually, the virus shedding load just hit the detection limit, and more cells would be passed at the other time, then the patient would have a re-positive test result. Of course, further precise and quantitative experimental studies are needed to verify this hypothesis.

It is crucial to clarify the infectivity in re-positive patients clinically and in public health decision making. However, it remains unknown despite substantive reports and research. The virus's ability to replicate in cultured cells serves as a surrogate marker of infectivity, but we did not perform virus culture due to technology limitations. On the other hand, culture is less sensitive than RT-PCR for detection of live viruses. Even with the culture of virus, mixed results were reported in different studies. Wölfel R and colleagues (*2020*)found that no virus could be cultured in patient samples after day 9 of symptom onset. In contrast, Liu WD and colleagues reported isolated SARS-CoV-2 from the sputum sample on day 18 after symptoms onset (*Liu et al., 2020*). Therefore, there is still no applicable clinical method to establish SARS-CoV-2's infectivity. The potential risk of virus transmission from a re-positive patient cannot be eliminated.

In Macau, most of our citizens are susceptible to COVID-19 since there is no local epidemic, and only very few residents were infected. Fortunately, enough resources such as AIIR, PPE, healthcare human resources, etc., which allow high quality and prolonged quarantine and minimize transmission risk in our city. After balancing the risk and benefit, we released re-positive patients from quarantine after three consecutive negative results. Zou Y and colleagues also supported that patients with three consecutive negative results had a much lower recurrent positive rate than two negative findings (*Zou et al., 2020*).This public health strategy contributes to Macau's achievements of beating back COVID-19. However, it did bring some undesired experience to the individuals. For example, prolonged isolation in a health facility is prone to psychological problems. As a result, other cities may not simply copy our methods. The health authority should make their decision according to community acceptance ability. Nonetheless, we agreed that quarantine at home for two weeks with PCR tests after hospital discharge is the least transmission prevention

since SARS-CoV-2 is a novel virus with only scanty knowledge about it and no effective treatment until now.

Lastly, some limitations of this study should be noted. First, the sample size was relatively small. There were only 35 cases in our study, even though all patients in Macau's second outbreak had been included. Second, the onset time and the presentation of sign or symptom was according to the patient's statement so that recall bias may present in the timeline of disease progression. Third, we did not perform the virus culture, and the infectivity could not be shown. Fourth, we could not get Ct values and genomic sequencing of SARS-CoV-2 due to laboratory policy. They should be considered in future studies.

According to the global COVID-19 situation, the disease seems to keep coexisting with humans for a long time. Since a high re-positive rate among COVID-19 patients was presented in our study, we can imagine that there will be a great practical challenge to handle this group of patients. Different underlying causes had different approaches. How can we distinguish re-infection, reactivation, and residual virus nucleic acid from each other? To do or not to do the whole-genome sequencing? This is a question. For the patients with residual virus nucleic acid, overtreatment may do some harm to individuals.

## CONCLUSIONS

A re-positive SARS-CoV-2 test after having fulfilled discharge criteria is a common phenomenon. The patients diagnosed in the early stage of illness and those with extended hospitalization were more likely to have recurrent positive results. Virus re-infection is impossible in our report, and reactivation is also unlikely, so that residual virus nucleic acid should be a reasonable explanation for this phenomenon. As there is insufficient evidence to exclude infectivity in re-positive patients, it is better to quarantine patients after discharge for a prolonged period with repeated RT-PCR tests. It may minimize the community's potential risk, particularly in regions with relatively plenty of resources and low community infection rates, such as Macau.

### Funding
The authors received no funding for this work.

### Competing Interests
The authors declare there are no competing interests.

### Author Contributions
- Chi Leong Wong and Sao Kuan Lei conceived and designed the experiments, performed the experiments, analyzed the data, prepared figures and/or tables, authored or reviewed drafts of the paper, and approved the final draft.
- Chin Ion Lei, Iek Long Lo, Chong Lam and Iek Hou Leong conceived and designed the experiments, analyzed the data, authored or reviewed drafts of the paper, and approved the final draft.
## Human Ethics

The following information was supplied relating to ethical approvals (i.e., approving body and any reference numbers):

The Medical Ethical Committee of Centro Hospitalar Conde de São Januário, Macau SAR, China, granted Ethical approval to carry out the study within its facilities (Ethical Application Ref: 0044/MEC/N/2020).

## Data Availability

Raw measurements are available as a Supplementary File.

## Supplemental Information

Supplemental information for this article can be found online at http://dx.doi.org/10.7717/peerj.11170#supplemental-information.

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
