# Peer review of "Re-positive of SARS-CoV-2 test is common in COVID-19 patients after hospital discharge. Data from high standard post-discharge quarantined patients in Macao SAR, China"

_PeerJ, doi:10.7717/peerj.11170_

## Round 0.1 · original submission · Minor Revisions

Thank you for the manuscript updates. However both the reviewers have some remarks on the text demanding revisions. The topic is important and even might be considered controversial. Please check suggestions from reviewer #2. Please add recent references into discussion.

·

Basic reporting

I have no objection to this section.

Experimental design

I have no objection to this section.

Validity of the findings

Without exception, all the given values ​​of the statistical significance p of frequencies were made with errors, however, the results of correct calculations do not contradict the conclusions of this article.
The correct values of p ​​are given in my letter as a reviewer.

There are no comments or comments on other points in this section.

Additional comments

1) The significance p of the frequency differences in the tables below is calculated incorrectly and should be corrected. The correct values ​​are given in my reviewer letter.
2) This article gives the frequency of long-term presence of the pathogen COVID-19, which is very different from the estimates, published in other articles. It might also be appropriate to discuss laboratory diagnostic errors as a possible reason for such unexpected results.

·

Basic reporting

Basic Reporting criteria generally fulfilled. A few grammatical errors were spotted (eg. "...before they (are) back to the Macau community."). Tenses should be past whenever appropriate (eg. "Here we reported the re-positive rate..." instead of "Here we will report the re-positive rate..."). I suggest authors to do a checking on these before re-submission.

I found the link to Ref 14 is dead. I suggest authors to do a checking on these before re-submission.

Experimental design

This report is a case-series of re-positive COVID19 cases in Macau, which fulfilled aims and scope of the journal. Ethical requirements were fulfilled.

I understand from the Guidelines cited in Ref 14 that 2 negative results of RT-PCR are required for discharge. This should be emphasized in the manuscript.

Having Ct values of the 12 re-positive cases would add value to this paper.

Validity of the findings

What do the authors mean by "...residual viruses in vivo cells."? If this means living cells, the authors themselves admitted that no cultures were taken to prove living virus, and subsequently infectivity. I strongly suggest modifications to all statements in the manuscript that may lead to proposition that the re-positive cases demonstrated nucleic acid amplification from living virus.

I think the findings are valid, within its limitations. The manuscript has added to previous reports of re-positive COVID19 cases post-discharge, without providing substantial additional impact/novelty.

---

## Round 0.2 · accepted · Accept

The reviewers have no critical remarks now.

·

Basic reporting

All my comments on this article, expressed in the first review, have been completely eliminated.

Experimental design

All my comments on this article, expressed in the first review, have been completely eliminated.

Validity of the findings

All my comments on this article, expressed in the first review, have been completely eliminated.

Additional comments

I read your article with great interest. I am sure that others will find it very interesting and useful.

·

Basic reporting

Acceptable. Concerns on grammatical errors and dead link were addressed adequately.

Experimental design

Appropriate. Concern adequately addressed.

Validity of the findings

Appropriate. Concerns addressed.

Additional comments

No further comment.